evolution/genetics/bioinformatics

genetic diversity, Nubian ibex, cytochrome *b*, D-loop, Oman, conservation

**Author for correspondence:**
Mataab K. Al-Ghafri
e-mail: alghafrimksa@yahoo.com

# Genetic diversity of the Nubian ibex in Oman as revealed by mitochondrial DNA

Mataab K. Al-Ghafri[1,2,3], Patrick J. C. White[1], Robert A. Briers[1], Kara L. Dicks[2], Alex Ball[2], Muhammad Ghazali[2], Steven Ross[3], Taimur Al-Said[3], Haitham Al-Amri[3], Mudhafar Al-Umairi[3], Hani Al-Saadi[3], Ali Aka'ak[3], Ahmed Hardan[3], Nasser Zabanoot[3], Mark Craig[4] and Helen Senn[2]

[1]School of Applied Sciences, Edinburgh Napier University, Edinburgh, UK
[2]RZSS WildGenes, Royal Zoological Society of Scotland, Edinburgh, UK
[3]Office for the Conservation of Environment, Diwan of Royal Court, Muscat, Oman
[4]Al Ain Zoo, Life Sciences Department, Abu Dhabi, United Arab Emirates

MKA-G, 0000-0002-4918-1530; PJCW, 0000-0002-9349-8447;
RAB, 0000-0003-0341-1203; KLD, 0000-0002-3764-4315;
AB, 0000-0003-1186-2717; HS, 0000-0002-3711-8753

The Nubian ibex (*Capra nubiana*) is patchily distributed across parts of Africa and Arabia. In Oman, it is one of the few free-ranging wild mammals found in the central and southern regions. Its population is declining due to habitat degradation, human expansion, poaching and fragmentation. Here, we investigated the population's genetic diversity using mitochondrial DNA (D-loop 186 bp and cytochrome *b* 487 bp). We found that the Nubian ibex in the southern region of Oman was more diverse (D-loop HD; 0.838) compared with the central region (0.511) and gene flow between them was restricted. We compared the genetic profiles of wild Nubian ibex from Oman with captive ibex. A Bayesian phylogenetic tree showed that wild Nubian ibex form a distinct clade independent from captive animals. This divergence was supported by high mean distances (D-loop 0.126, cytochrome *b* 0.0528) and high $F_{ST}$ statistics (D-loop 0.725, cytochrome *b* 0.968). These results indicate that captive ibex are highly unlikely to have originated from the wild population in Oman and the considerable divergence suggests that the wild population in Oman should be treated as a distinct taxonomic unit. Further nuclear genetic work will be required to fully elucidate the degree of global taxonomic divergence of Nubian ibex populations.

# 1. Introduction

The *Capra* (or goat) genus is distributed widely in the three continents of Europe, Africa and Asia, and its range extends from the cooler areas of the Alpine mountains to the hot hyper-arid desert of Arabia [1]. The number of species of *Capra* is debated and described as containing between six and nine species, but they are all rocky montane specialists [1,2]. According to phenotypic characteristics, *Capra* species are divided into three groups: markhor, ibex and true goats [3]. Heptner *et al.* [4] divided the ibex into seven species: Spanish ibex (*C. pyrenaica*), Alpine ibex (*C. ibex*), Dagestan tur (*C. cylindricornis*), Caucasian ibex (*C. caucasica*), Siberian ibex (*C. sibirica*), Nubian ibex (*C. nubiana*) and Walie ibex (*C. walie*). The more widely accepted classification, which is used by the International Union for the Conservation of Nature (IUCN), is that the *Capra* consists of nine species: the seven ibex, a single species of markhor (*C. falconeri*) and wild goat (*C. aegagrus*) (see electronic supplementary material, S1).

The Nubian ibex (*C. nubiana*) is the smallest *Capra* species [3]. Males weigh between 55 and 65 kg and are distinguished by long curved horns, while females are much lighter, weighing in the region of 21–27 kg with smaller and thinner horns [5,6] (figure 1*a*). The distribution of the Nubian ibex extends from northeast Africa through the Middle East and into the Arabian Peninsula [8] (figure 1*b*).

The taxonomic status of the Nubian ibex has been debated, with earlier studies tending to classify it as a subspecies of the Alpine ibex (*C. ibex*) because of the close similarity in the morphology of the horns [2]. Early comparisons used allozymes to compare the Nubian ibex with the Alpine ibex, but did not find enough supporting evidence to consider the Nubian ibex as a separate species [9,10]. On the other hand, a number of more recent studies have investigated the taxonomy of the genus *Capra* and classified the Nubian ibex as a separate species according to allozyme and mitochondrial DNA results [11–15], summarized in table 1.

Groves & Grubb [3] recommended further splitting of the Nubian ibex into two subspecies based on coat colour differences: the Sudanese Nubian ibex as *C. nubiana nubiana* (F. Cuvier, 1825), and those from the Dead Sea and Sinai as *C. nubiana sinaitica* (Ehrenberg, 1833). Groves & Grubb [3] debated the merit of species or subspecies classification of the Nubian ibex, as well as postulating a third subspecies in the Arabian Peninsula. However, a lack of specimens from southern Arabia has hindered the resolution of this debate and prevented further understanding of population substructure.

Delimiting the boundaries between species is vital for informing management decisions in a conservation context. The IUCN, generally recognized as authoritative for the purposes of conservation, currently classifies the Nubian ibex as a single species with a Red List classification of Vulnerable [7]. Here, we reopen the debate by studying samples from the most southeasterly region of the Nubian ibex's range, Oman.

The Nubian ibex is a flagship species for conservation efforts in Oman, alongside other important arid-land ungulates [17,18]. The species is located in fragmented populations from the central region down to the southern region of Oman [7,8,18] (figure 1*c*). The Nubian ibex in the central region is restricted to the 100–150 km long Al Wusta Wildlife Reserve (WWR) escarpment, a hyper-arid region [5]. The southern region, in contrast, is ecologically distinct from the central region, being higher and wetter, with the highest floral and faunal diversity of anywhere in Oman [19]. Throughout Oman, Nubian ibex populations are declining in response to poaching, human settlement expansion, feral livestock competition, habitat degradation and population fragmentation [17].

Perhaps surprisingly, there has so far been little genetic research on the Nubian ibex across its range, and this is especially lacking in the Arabian Peninsula. In Oman, there is a pressing need to design and inform a conservation management plan for the species. Therefore, we aim to (i) investigate the genetic diversity of the Nubian ibex in Oman and (ii) address the question of whether animals from captive populations would be suitable for future reintroduction/reinforcement programmes.

# 2. Material and method

## 2.1. The study area

The study areas consisted of three locations in Oman where the Nubian ibex is confirmed to exist: Al Wusta Wildlife Reserve (WWR), Shalim and Dhofar. WWR is a protected area in the hyper-arid central region (19.719960° N, 57.496767° E). The WWR sampling area (approx. 150 km long) included the southern part of the reserve, where the Nubian ibex is free ranging, and extended beyond the protected area boundaries to cover the largest area of the Nubian ibex range within the central region (figure 1*c*). The

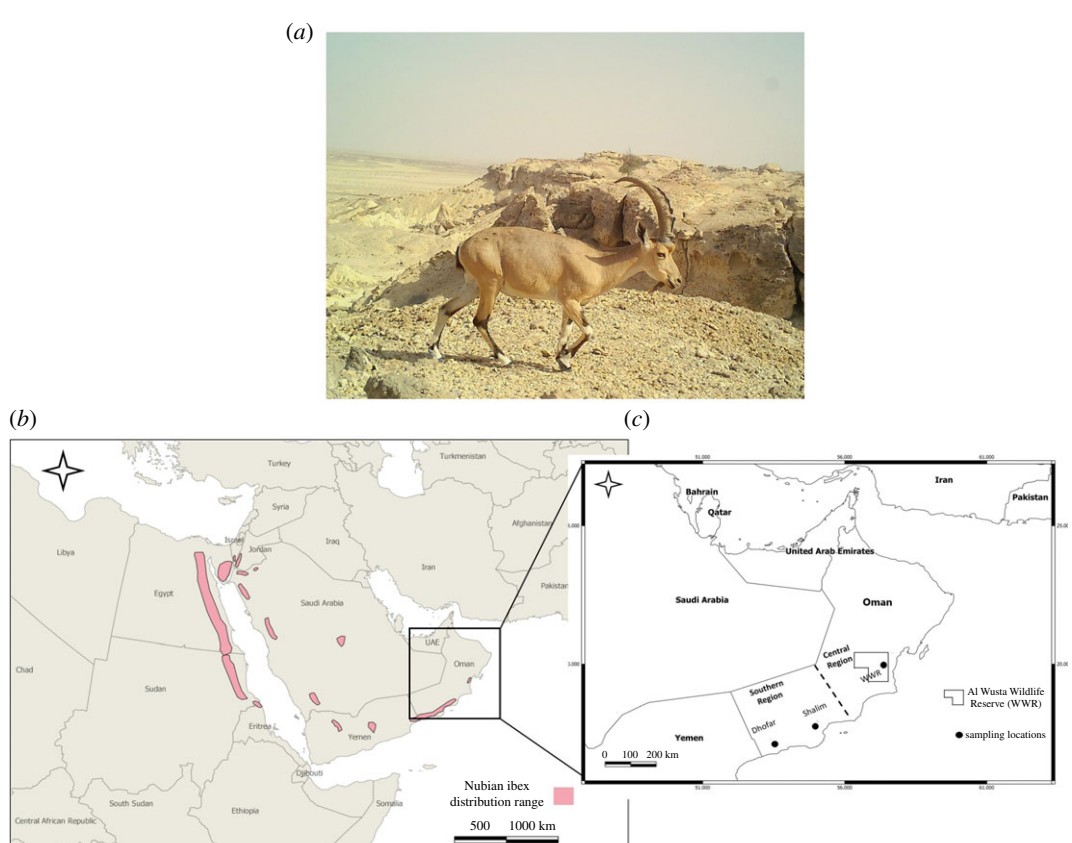

**Figure 1.** (*a*) Nubian ibex male captured by camera trap at Al Wusta Wildlife Reserve (WWR) (Oman) (by S.R. and T.A.-S.). (*b*) Distribution range of *C. nubiana* in the Middle East and Africa [7]. (*c*) Map of Oman showing the sampling locations represented by black circles. The samples were collected from three different places: Al Wusta Wildlife Reserve (WWR) and surrounds, Shalim and Dhofar.

**Table 1.** Published genetic studies investigating taxonomy of the *Capra*, highlighting their findings regarding the relationship between Nubian and Alpine ibex.

| reference | study species | marker used | Nubian ibex found to be a distinct species from Alpine ibex |
|---|---|---|---|
| Granjon *et al.* [10] | Alpine ibex; Nubian ibex | allozymes | no |
| Hartl *et al.* [15] | seven species of Caprini, and two species of Rupicaprini | | yes |
| Stüwe *et al.* [9] | Alpine ibex and Nubian ibex | | no |
| Manceau *et al.* [13] | eight *Capra* species | mitochondrial DNA (cytochrome *b* and D-loop) | yes |
| Ropiquet & Hassanin [16] | 17 Caprini species | mitochondrial DNA (12S, CO2, Cytb, ND1) and nuclear genes (kCas, PRKC1, SPTBN1 and TG) | yes |
| Pidancier *et al.* [14] | eight *Capra* species | mitochondrial DNA (cytochrome *b*) and nuclear Y-chromosome | yes |
| Kazanskaya *et al.* [12] | eight *Capra* species | mitochondrial DNA (cytochrome *b* and D-loop) | yes |

reserve contains no permanent human settlements, but is used frequently by nomadic pastoralists to herd camels and domestic goats. The second location was Shalim (18.107289° N, 55.650218° E), which is approximately 350 km southwest of WWR in the southern region (figure 1*c*). Finally, Dhofar is

**Table 2.** Locations and types of samples collected for analysis.

| population | sampling location/source | collection date | faecal | blood | tissue | bone/horn | total |
|---|---|---|---|---|---|---|---|
| wild Oman (central region) | Al Wusta Wildlife Reserve (WWR) | 2014–2018 | 55 | 1 + 1[a] | 3[a] | 12 | 71 |
| | WWR (Natural History Museum Oman) | 2019 | 0 | 0 | 1 | 7 | 8 |
| wild Oman (southern region) | Shalim | 2013–2018 | 0 | 0 | 14 | 0 | 14 |
| | Shalim (Natural History Museum Oman) | 2019 | 0 | 0 | 1 | 1 | 2 |
| | Dhofar | 2019 | 84 | 0 | 0 | 3 | 87 |
| Oman captive | Bait Al Barakah Breeding Centre (Muscat, Oman) | 2017 | 0 | 9 | 0 | 0 | 9 |
| UAE captive | Al Mayya Sanctuary (Fujairah, United Arab Emirates) and Al Ain Zoo (Abu Dhabi, United Arab Emirates) | 2015–2018 | 0 | 56 | 0 | 0 | 56 |

[a]One individual has two samples (one tissue and one blood).

approximately 200 km southwest of Shalim. Four locations throughout Dhofar were sampled because, it was expected to host a large population of Nubian ibex compared with WWR and Shalim.

## 2.2. Sampling

### 2.2.1. Wild samples

As a part of the Nubian ibex conservation programme launched by the Omani Office for Conservation of the Environment in 2014, 55 faecal samples were collected non-invasively during setting of camera traps in WWR. Additionally, 12 bone samples and two dried tissue samples were collected from skulls and horns of deceased animals. A single tissue sample was collected from each of six hunted animals, which were confiscated by Royal Oman Police (ROP) in January 2018 in the Shalim area [20]. Eight tissue samples from hunted ibex from the same area were also sent from Oman to the Royal Zoological Society of Scotland (RZSS) in 2015, but it was not recorded whether the number of samples represents eight individuals or fewer (i.e. multiple samples may have been taken from the same animal(s)).

From Dhofar, 84 faecal samples were collected non-invasively and three bone/horn samples were collected from deceased animals. Additional samples were collected from the Natural History Museum of Oman, Muscat, which represent Nubian ibex specimens collected from the wild during survey studies. These included eight samples from WWR and two samples from Shalim.

### 2.2.2. Captive samples

Nine blood samples were collected from animals at Bait Al Barakah breeding centre (the Royal private collection in Muscat, Oman) as part of routine veterinary examinations of captive-bred animals (table 2). A further 56 blood samples of Nubian ibex from UAE captive populations were used with permission from Al Mayya Sanctuary and Al Ain Zoo in the United Arab Emirates. Faecal samples from the wild and blood samples from captive centres were collected between 2014 and 2018 (table 2). A full list of samples is found in electronic supplementary material, table S2.1.

## 2.3. DNA extraction

DNA from blood and tissue samples was extracted using the DNeasy Blood and Tissue kit (QIAGEN®, Germany) according to the manufacturer's instructions. For bone samples, DNA was extracted using

**Table 3.** Primer sequences used to amplify targeted markers. All primers were designed for the purposes of this study.

| marker | primer ID | primer sequence (5'>3') | estimated fragment length |
|---|---|---|---|
| D-loop | NUB_D-loop_RZSS_F1 | ATGGCACTAATGCAACAAG | 242 bp |
| | NUB_D-loop_RZSS_R1 | TGCTATGTACGGGTATGCAG | |
| cytochrome *b* | NUB_CytB_RZSS_F1 | GGACGAGGCCTATATTATGGA | 244 bp |
| | NUB_CytB_RZSS_R1 | CGAAAAATCGGGTGAGAGTG | |
| | NUB_CytB_RZSS_F2 | TTGGCACAAACCTAGTCGAA | 282 bp |
| | NUB_CytB_RZSS_R2 | GCAGGTCGGGTGTGAATAGT | |
| | NUB_CytB_RZSS_F3 | CTGCTCTTCCTCCACGAAAC | 369 bp |
| | NUB_CytB_RZSS_R3 | TGGGCGGAATATTATGCTTC | |

QIAamp DNA Investigator Kit (QIAGEN®) according to the manufacturer's bones and teeth protocol. DNA was extracted from the faecal samples using the Isohelix Xtreme DNA Kit (XME-50); see the electronic supplementary material, S3 for details.

## 2.4. Mitochondrial DNA sequencing

Primer pairs for both D-loop and cytochrome *b* were designed with Primer3Plus [21], using a GenBank reference sequence from each of the different *Capra* species in order to identify conserved regions (accession numbers included in electronic supplementary material, S4). Because the D-loop is highly variable and the DNA from the faecal and bone samples was expected to be fragmented due to degradation over time [22], one hyper-variable segment (242 bp) was targeted and amplified using a single primer pair. For cytochrome *b*, three primer pairs were designed to amplify short, overlapping fragments. Table 3 shows the primer sequences used for the D-loop and cytochrome *b*.

PCRs were carried out at a total volume of 10 µl and contained 1 µM of each primer, 1.4X DreamTaq Hot Start Master Mix (Thermo Fisher Scientific Inc., includes 0.28 mM of each dNTP and 2.8 mM MgCl$_2$) and 1 µl of extracted DNA. For the faecal samples, 0.2 µl of bovine serum albumin (BSA) were added to the master mix. The PCR programme for both D-loop and cytochrome *b* was as follows. The initial denaturation step was at 95°C for 5 min. This was followed by 39 cycles of 95°C denaturation for 30 s, 55°C for 30 s to allow primer annealing and then 72°C for 60 s for elongation. A final 72°C extension for 10 min completed the programme. PCR products were confirmed by gel electrophoresis before being cleaned up with 0.5 µl of EXO1 enzyme and 0.5 µl FastAP. PCR products were Sanger sequenced in both directions using the Big Dye Terminator Kit v. 3.1 (Applied Biosystems) on an ABI 3730 DNA Analyzer (Applied Biosystems).

## 2.5. Sequence analysis

Sanger sequences were trimmed and quality checked by eye using Geneious software (v. 11.1.5). The three overlapping cytochrome *b* sequences generated from the same sample were aligned and the consensus sequence generated. The good quality sequences were then aligned with reference sequences from GenBank (see electronic supplementary material, S4) and trimmed to an equal size of 186 bp (D-loop) and 486 bp (cytochrome *b*) using MEGA X [23].

To assess the genetic diversity of the Nubian ibex from Oman in relation to individuals from other sources, median-joining [24] and TCS haplotype networks were built using PopArt software v. 1.7 [25] for both the D-loop and cytochrome *b*. To estimate the robustness of the sample sizes obtained by this study, we produced haplotype accumulation curves using the R (v. 3.5.3) package HACSim [26–28]. This is used to estimate the total sample size, which is required to capture all the haplotypes in a specific population.

The accession numbers for other *Capra* species used to construct the phylogenetic trees can be found in the electronic supplementary materials, S4. The concatenation of D-loop and cytochrome *b* to a total length of 673 bp was performed using Geneious (v. 11.1.5). In addition, the cytochrome *b* sequences were translated into amino acids to check for evidence of incorrect amplification of nuclear insertions of mitochondrial sequences (NuMtS) [29].

## 2.6. Phylogenetic analysis

A phylogenetic tree was constructed for the concatenated sequences. Whole mitochondrial genome sequences for different *Capra* species along with other species sequences imported from GenBank were used and can be found in the electronic supplementary material, S3. *Bos taurus* (AY676870) was used as an outgroup. The evolutionary model used for conducting the analyses was HKY + $\Gamma$ + I, as selected using jModelTest in the R package 'phangorn' [30]. The phylogenetic tree for the concatenated sequences was constructed by MrBayes [31] within Geneious (v. 11.1.5). The parameters used were as follows: total chain length 1 000 000, subsample frequency 200 and a burn-in of 10% of the trees was applied.

## 2.7. Genetic diversity statistics

The mean genetic distances between the wild Nubian ibex samples from each location in Oman and the captive animals were calculated with MEGA X using the maximum composite likelihood [23,32]. In addition, the genetic distance and differentiation between the wild Nubian ibex and the captive ibex were calculated using DnaSP v. 6.12.03 [33]. Analysis of molecular variance (AMOVA) was carried out using Arlequin v. 3.5 [34]. This was used to measure the population genetic structure within and between groups, using 159 and 131 sequences for D-loop and cytochrome $b$, respectively. The permutation was set to default (1023) at a significance value of $p = 0.05$.

# 3. Results

## 3.1. Haplotype networks

A total of 188 sequences were successfully amplified for the cytochrome $b$. The quality control check excluded 57 samples due to poor amplification. Therefore, the final sample set included 131 high-quality sequences, which were used to create an alignment with total length of 487 bp for the analysis (see electronic supplementary material, S2.2). A total of seven cytochrome $b$ haplotypes were found in this study (named, A–G). Wild Nubian ibex from Oman were found to have three haplotypes (A, B and C), while the captive populations were found to contain four haplotypes (D, E, F and G). There were no shared haplotypes between the wild and captive populations. Regarding the wild sampling locations in Oman, two haplotypes were identified in the WWR (A and B). Dhofar was found to share both these haplotypes, while Shalim only had one haplotype (A, shared with both WWR and Dhofar). Dhofar, on the other hand, had an additional unique haplotype (C). The geographical distribution of the haplotypes of the wild Nubian ibex in Oman is illustrated in figure 2.

The Omani captive population had three cytochrome $b$ haplotypes in total, including two unique haplotypes (D and G) and one haplotype shared with the UAE captive population (E). In addition, the UAE captive population had an additional haplotype (F; figure 3) (electronic supplementary material, S2.3). All haplotype sequences were submitted to NCBI GenBank under accession numbers (MW911255–MW911278).

A total of 159 sequences were successfully amplified for the D-loop and were used in the analysis. The total sequence alignment length was 186 bp, revealing 17 haplotypes (numbered 1–17). There were no haplotypes shared between the wild and captive Nubian ibex. The wild Nubian ibex exhibited 12 D-loop haplotypes (1–12), while the captive ibex have five different haplotypes (13–17). WWR was found to have three D-loop haplotypes (1–3), all of which were shared with Shalim. Shalim had six D-loop haplotypes, three of which were unique (4, 5 and 8). Dhofar had six unique D-loop haplotypes (6, 7, 9, 10, 11 and 12), which were not shared with either Shalim or the WWR. On the other hand, the Omani captive animals were found to have three D-loop haplotypes, two of which were unique and one which was shared with the UAE captive animals. The UAE sample also had two unique D-loop haplotypes not shared with Omani captive animals (figure 4).

The median-joining networks showed a clustering pattern for the wild Nubian ibex in both the cytochrome $b$ (figure 3) and the D-loop (figure 4). There are 21 mutations separating the wild from the captive ibex in cytochrome $b$ and 12 mutations in D-loop.

For cytochrome $b$, the haplotype accumulation curve reached an asymptote (figure 5) where 98.5% of the haplotypes have been sampled at $p = 0.05$ confidence. On the other hand, the haplotype accumulation curve for the D-loop was slightly below the asymptote, but the difference between the sampled and

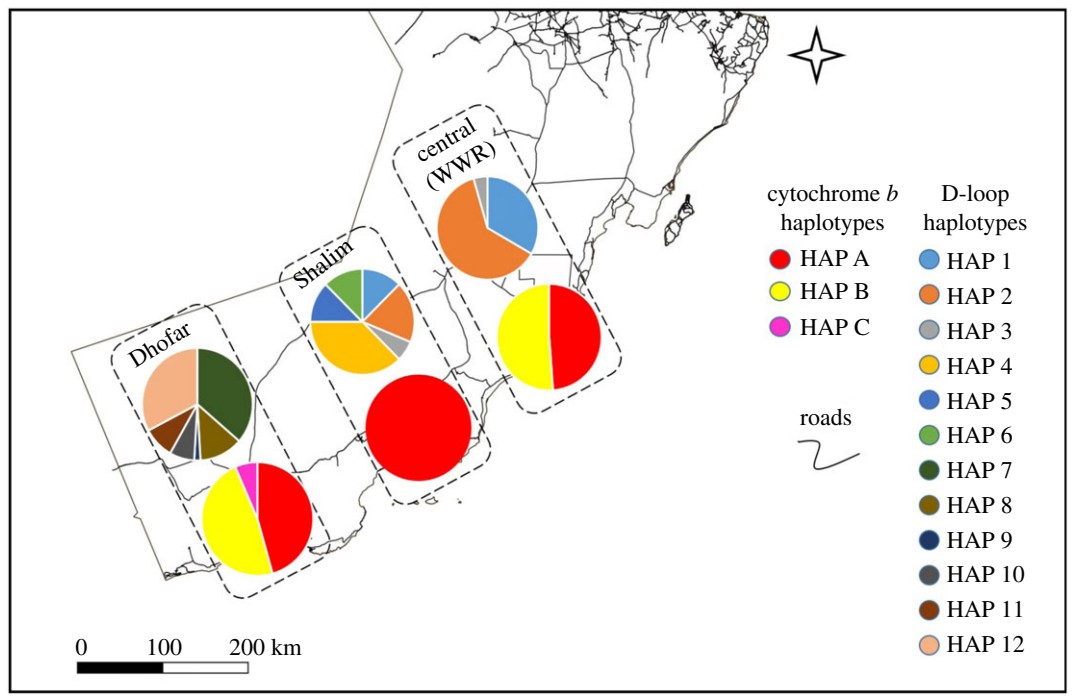

**Figure 2.** The distribution of the mitochondrial DNA haplotypes within each location. The lower circle for each region represents cytochrome *b*, while the upper circle is D-loop. The size of the circle does not represent the sample size.

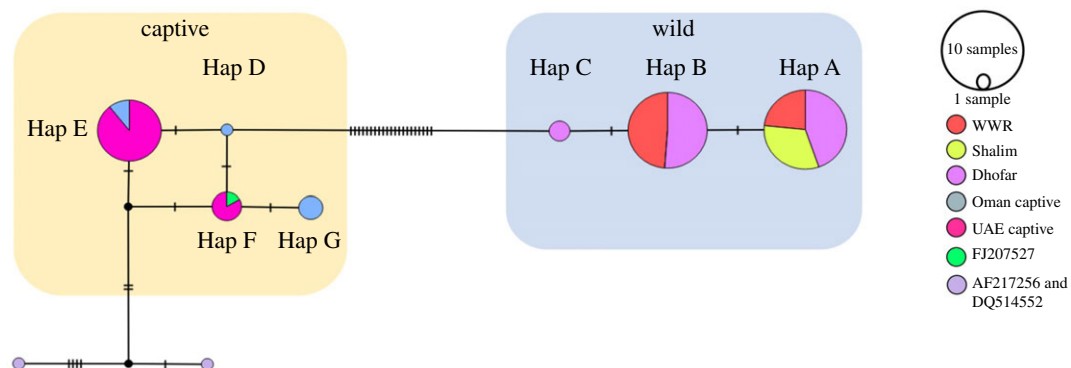

**Figure 3.** Median-joining haplotype network for cytochrome *b* (487 bp). The wild animals from Oman are in the right-hand blue box, while the captive animals are in the left-hand orange box. Each coloured circle represents a single haplotype, the size of which is proportional to the number of samples. The tick marks represent the mutational steps between haplotypes. Three reference sequences of *C. nubiana* from GenBank were used (accession numbers given on the figure).

unsampled haplotypes is small (*R* = 93.7% of the haplotypes were sampled at a 95% confidence interval) (figure 5). This indicates that the sampling process is likely to be representative of the Nubian ibex populations in Oman.

## 3.2. Phylogenetic trees

The number of concatenated mitochondrial DNA (mtDNA) haplotypes (cytochrome *b* and D-loop) was 18. The wild Nubian ibex from Oman had 13 concatenated haplotypes (named WildHAP1 to WildHAP13). The captive animals had five concatenated haplotypes, named (CaptiveHAP14 to CaptiveHAP18).

The phylogenetic tree of the concatenated mtDNA sequences shows separation of the *C. nubiana* from the rest of the *Capra* species (figure 6). This separation is supported with a 0.99 posterior probability. Our data indicate two well-supported clades within *C. nubiana*: the first one contained the wild samples from Oman, while the second one contains the samples from Oman and UAE captive animals. The divergence

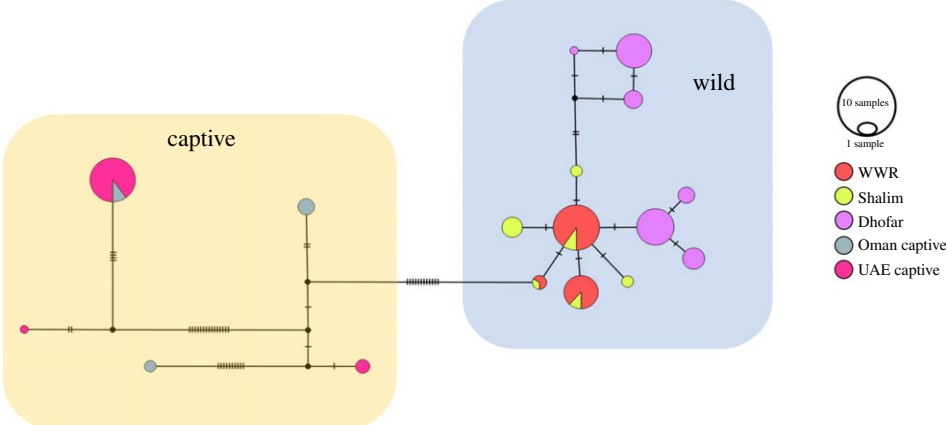

**Figure 4.** Median-joining haplotype network for D-loop (186 bp). The wild animals from Oman are in the right-hand blue box, while the captive animals are in the left-hand orange box. Each coloured circle represents a single haplotype, the size of which is proportional to the number of samples. The tick marks represent the mutational steps between haplotypes.

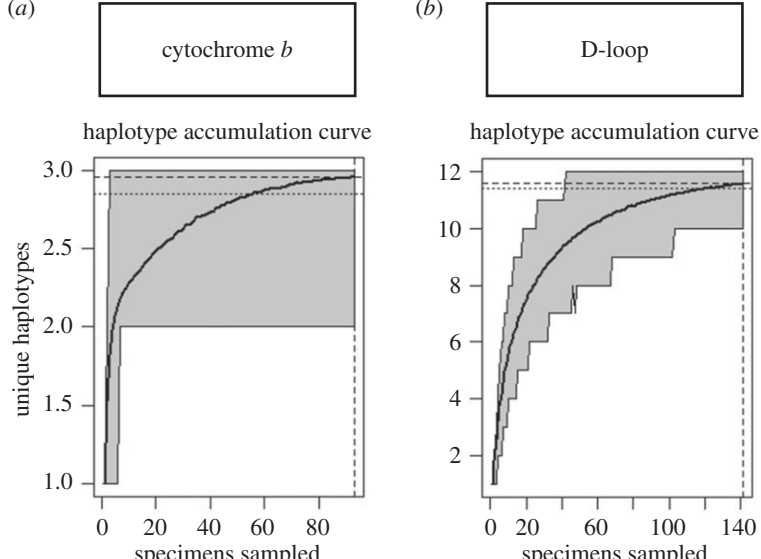

**Figure 5.** Haplotype accumulation curves for cytochrome *b* (*a*) and D-loop (*b*) within the wild Nubian ibex samples. For cytochrome *b*, it is estimated that more than 98.5% of the haplotypes are represented, while for the D-loop, this is 93.7%. The dashed lines show the number of haplotypes found corresponding with sampled individuals. The dotted lines represent the expected numbers of haplotypes which should be found in the given population.

is supported by a posterior probability of 1.0 (figure 6). The concatenated haplotypes from captive samples cluster with the reference sequences from GenBank (NC020624 and FJ207527) [35]. This suggests that the original source of the captive animals in both Oman and the UAE is similar, but was not from the current distribution of the wild Omani population. Unfortunately, the original collection locations of the GenBank sequences (museum samples) were not detailed.

Additional trees, which were constructed using Bayesian analysis within StarBEAST 2 software [36], retrieved the same tree topology ensuring the stability of the phylogenies (see electronic supplementary material, S5). The Bayesian phylogeny shows a large evolutionary separation between the wild Nubian ibex in Oman and the captive Nubian ibex samples.

## 3.3. Nubian ibex genetic diversity

Both cytochrome *b* and D-loop had higher haplotype diversity in the wild Nubian ibex (0.54 and 0.85, respectively) compared with the captive ibex (0.44 and 0.47, respectively). However, in both regions,

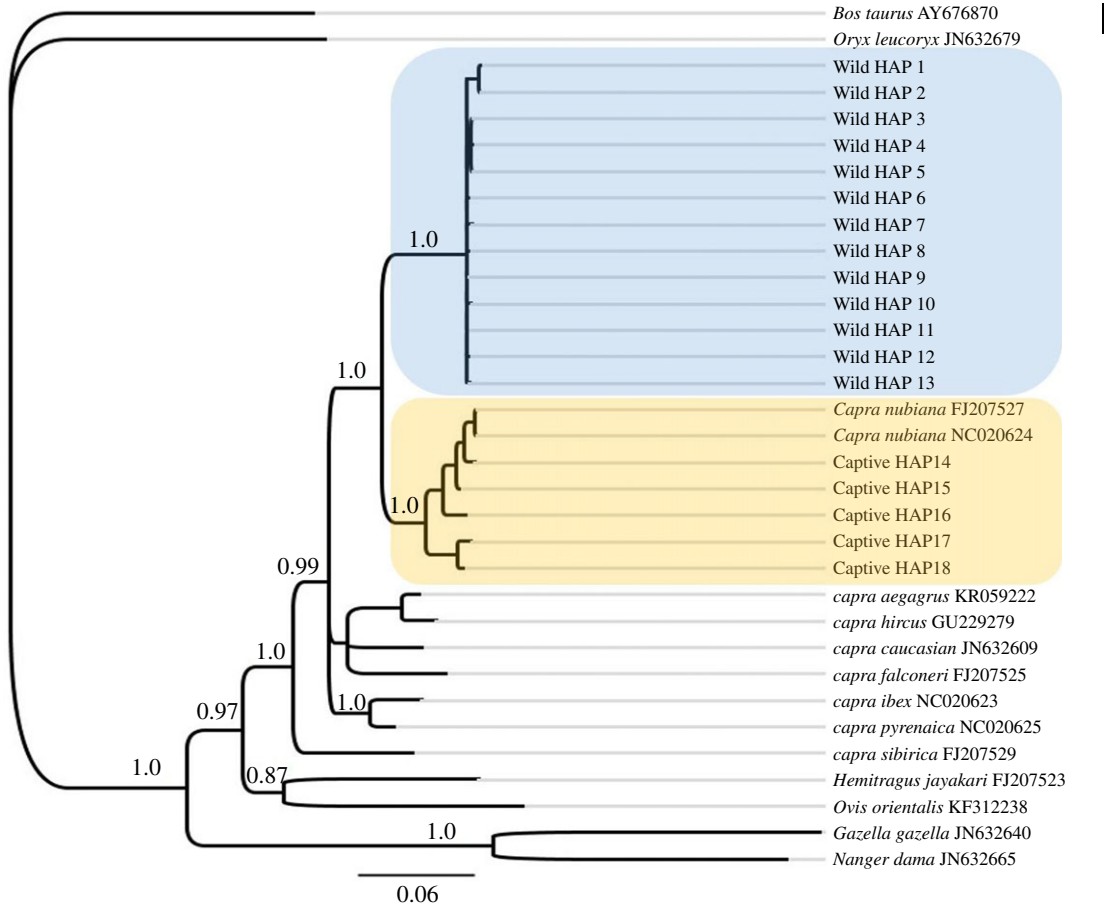

**Figure 6.** Bayesian phylogenetic tree of concatenated mtDNA sequences of cytochrome *b* (487 bp) and D-loop (187 bp) calculated by MrBayes. The blue box indicates Oman wild Nubian ibex (WildHAP1–WildHAP13). The yellow box indicates the captive Nubian ibex (CaptiveHAP14–CaptiveHAP18). GenBank sequences are indicated by their accession numbers adjacent to the species binomial. The numbers at the nodes represent the posterior probabilities.

**Table 4.** Molecular diversity measures for both cytochrome *b* and D-loop between wild and captive animals.

| population | cytochrome *b* | | D-loop | |
|---|---|---|---|---|
| | Oman wild | captive | Oman wild | captive |
| number of haplotypes, *h* | 3 | 4 | 12 | 5 |
| haplotype diversity, Hd | 0.54 | 0.44 | 0.85 | 0.47 |
| segregating sites, *S* | 2 | 3 | 12 | 32 |
| nucleotide diversity, Pi | 0.001 | 0.002 | 0.014 | 0.046 |
| number of sequences | 93 | 38 | 116 | 43 |
| sequence length | 487 bp | | 186 bp | |

nucleotide diversity was higher in the captive ibex (0.002 and 0.046) compared with the wild Nubian ibex (0.001 and 0.014). Table 4 shows the molecular diversity measures of the wild and captive Nubian ibex.

Maximum-likelihood mean distances between wild Nubian ibex populations in Oman and the captive animals for cytochrome *b* and D-loop can be found in electronic supplementary material, table S6.1.

In terms of the wild Nubian ibex, it is clear that the haplotypes and nucleotide diversity is higher in the southern region than in the central region (table 5). In addition, the $F_{ST}$ value between the central region and the southern region is 0.09 for cytochrome *b* and 0.32 for the D-loop.

**Table 5.** Comparison of genetic diversity indices between Nubian ibex samples from the central and southern region.

| | central region (WWR) | | southern region (Shalim & Dhofar) | |
|---|---|---|---|---|
| | cytochrome *b* | D-loop | cytochrome *b* | D-loop |
| number of haplotypes, *h* | 2 | 3 | 3 | 12 |
| haplotype diversity, Hd | 0.465 | 0.511 | 0.587 | 0.838 |
| segregation sites, *S* | 1 | 2 | 2 | 12 |
| nucleotide diversity, Pi | 0.001 | 0.002 | 0.0012 | 0.0177 |
| number of sequences | 32 | 45 | 61 | 71 |

**Table 6.** Comparison of genetic diversity indices for Nubian ibex from WWR, Shalim and Dhofar.

| | cytochrome *b* | | | D-loop | | |
|---|---|---|---|---|---|---|
| population | WWR | Shalim | Dhofar | WWR | Shalim | Dhofar |
| number of haplotypes, *h* | 2 | 1 | 3 | 3 | 6 | 6 |
| haplotype diversity, Hd | 0.47 | 0.0 | 0.57 | 0.51 | 0.83 | 0.65 |
| segregating sites, *S* | 1 | 0 | 2 | 2 | 5 | 9 |
| nucleotide diversity, Pi | 0.001 | 0.0 | 0.001 | 0.002 | 0.005 | 0.016 |
| number of sequences | 32 | 15 | 46 | 45 | 16 | 55 |
| sequence length | 487 bp | | | 186 bp | | |

Furthermore, the genetic diversity of the Nubian ibex in terms of the samples location showed that haplotype diversity is higher in Dhofar (0.57) for cytochrome *b* compared with WWR (0.47) (table 6). For the D-loop, Shalim has the highest haplotype diversity (0.83), while Dhofar is intermediate (0.65), and WWR has the lowest diversity (0.51) compared with other locations (table 6). Moreover, the $F_{ST}$ for D-loop between Dhofar and Shalim was found to be 0.39, while the $F_{ST}$ between Dhofar and WWR was 0.43 and between Shalim and WWR was 0.15. Therefore, the greatest similarity (and likely gene flow) is between Shalim and WWR.

## 3.4. Genetic divergence between wild and captive Nubian ibex

Pairwise differences showed that there was 0.053 (5.3%) divergence at cytochrome *b* and 0.126 (12.6%) at the D-loop between the wild and captive Nubian ibex. In addition, the $F_{ST}$ between the wild and captive populations calculated by DnaSP v. 6.0 was found to be 0.97 and 0.73 for cytochrome *b* and D-loop, respectively, indicating a large genetic difference between the wild and the captive animals. AMOVA results showed high diversity in cytochrome *b* among the Nubian ibex populations in Oman and the captive animals (97.19%, $p < 0.0001$; table 7). The D-loop statistics support these results, with the highest diversity among populations compared with within populations (77.87%, $p < 0.0001$). The significant diversity between wild Nubian ibex in Oman and the captive population is further supported by genetic distance values ($F_{ST}$ 0.97, $p < 0.0001$ and $F_{ST}$ 0.78, $p < 0.0001$) for cytochrome *b* and D-loop, respectively (table 7).

The mean divergence distance between the wild Oman and reference samples from Egypt (AJ009879) and Israel (DQ514552 and AF217256) within cytochrome *b* was found to be 0.072 (7.2%) and 0.055 (5.5%), respectively. Additional comparison between wild Oman and other *Capra* species can be found in electronic supplementary material, table S6.2.

## 4. Discussion

Previous studies on the Nubian ibex in the Arabian Peninsula have mostly focused on ecology and population distribution [5,6,37]. This is the first study to address the genetic diversity of the Nubian

**Table 7.** Analysis of molecular variance for cytochrome *b* and D-loop among Nubian ibex in the wild (Oman) and in captivity (Oman and UAE).

| D-loop among two populations (Omani wild and captive) | | | | | | |
|---|---|---|---|---|---|---|
| source of variation | d.f. | sum of square | variance components | percentage variation | fixation index (Fst) | *p*-value |
| among populations | 1 | 468.185 | 7.428 Va | 77.87 | 0.779 | <0.0001 |
| within populations | 157 | 331.368 | 2.111 Vb | 22.13 | | |
| total | 158 | 799.553 | 9.539 | | | |
| cytochrome *b* among two populations (Omani wild and captive) | | | | | | |
| among populations | 1 | 635.652 | 11.775 Va | 97.19 | 0.972 | <0.0001 |
| within populations | 129 | 43.966 | 0.341 Vb | 2.81 | | |
| total | 130 | 679.618 | 12.11583 | | | |

ibex in Oman. It provides the first insight into the population genetic structure of this species and produces vital baseline information for future management strategies.

Based on the results of mtDNA of 186 bp D-loop and 487 bp cytochrome *b* sequences, we found that the populations in the southern region (Shalim and Dhofar) are genetically more diverse than the central region (WWR). The haplotype accumulation curves indicate that the sampling process has provided a highly representative sample set that captures a good representation of haplotypes present in the Omani populations (figure 5). The gene flow between these populations is restricted ($F_{ST} = 0.32$ for the D-loop), although there may be greater gene flow between Shalim and WWR (evidenced by shared haplotypes and $F_{ST} = 0.15$ for D-loop). One cytochrome *b* haplotype is shared between the central and the southern region and three D-loop haplotypes are shared only between the central region and its closest population in the southern region, Shalim. This is likely to be attributed to a combination of a stepping-stone pattern [38] between the three populations, the semi-isolation of the central region population and the extremely limited connectivity between regions due to human activities such as road construction and oil exploration.

## 4.1. Population-level genetic substructure of the Nubian ibex in Oman

The results reveal genetic substructuring within Nubian ibex populations in Oman, clearly illustrated by the mtDNA haplotype distribution. Nubian ibex in Oman appear to display a stepping-stone pattern [38], where only neighbouring populations share haplotypes. The central region population (WWR) is in close proximity with the Shalim population in the southern region ($F_{ST} = 0.15$) which probably explains the shared D-loop haplotypes between them. The longer distance between the Dhofar population and both Shalim and the central population (WWR), resulting in very limited gene flow among the regions (Dhofar–Shalim $F_{ST} = 0.39$ and Dhofar–WWR $F_{ST} = 0.43$), probably explains the lack of shared haplotypes.

The genetic diversity of the Nubian ibex in WWR is remarkably lower than that of Dhofar. This low level of diversity may be attributed to the isolated nature of the population with restricted or even no gene flow, which could be due to human activities and, specifically, oil exploration or roads which bisect wildlife corridors. For example, Ross *et al.* [39] found two genetic clusters in the Arabian tahr (*Arabitragus jayakari*) population in Oman, which were separated by human barriers (i.e. roads and highways). They speculated that these roadblocks will contribute to increasing the genetic divergence of these populations and will eventually cause an increase in inbreeding in the long term. The same factors found by Ross *et al.* [39] may be applied to the Nubian ibex population in the central and southern regions. These factors include resistance to movements between the populations, the long distances between them, and urbanization and development. In addition, despite the substantial efforts taken by the authorities to limit and minimize hunting, wildlife populations (Arabian Oryx— *Oryx leucoryx*, Mountain gazelle—*Gazella gazella* and Nubian ibex) in the central region have suffered from high hunting pressure, especially between 1996 and 1999 [40].

The relatively high genetic diversity found in the Dhofar population could be associated with a relatively large and highly diverse area of habitat. In addition, the population estimated there is larger

than that of WWR, which ranged between 600 and 1100 individuals [7]. Furthermore, the possible connectivity with the Yemeni population could play a vital role in exchange of immigrants between the populations. However, it is not clear whether these populations are still connected given that many areas which could function as connections between the populations in Oman and Yemen are now occupied by human settlements or intersecting roads [17]. In addition, a security fence has been constructed between the borders of Oman and Yemen, which is likely to further limit any gene flow.

It should be noted that the analyses carried out here have used only mitochondrial DNA, which is only inherited maternally, and are, therefore, limited in their scope to elucidate population structure and gene flow in a species for which male-biased dispersal is typical. Further research using nuclear markers would provide valuable insight into whether gene flow between populations in Oman remains limited after accounting for male-biased dispersal. Nevertheless, further investigation is needed on anthropogenic effects (such as road building, hunting and oil exploration) on wildlife in Oman and how they affect avoidance behaviour, movement and distribution. Analyses involving nuclear markers may help to elucidate whether the extent of genetic substructuring of Nubian ibex populations in Oman is a result of historical or human-driven effects or both.

## 4.2. The genetic diversity of the Nubian ibex in Oman

In general, the D-loop haplotype diversity estimate for the wild Nubian ibex in Oman (Hd = 0.85) was found to be in the same range as those of other wildlife species, which indicates a considerable level of genetic diversity. On the other hand, the nucleotide diversity (Pi = 0.014) was low compared with other wild ungulate species. For example, haplotype and nucleotide diversity in dama gazelles (*Nanger dama*) from the wild population in central Chad was found to be Hd = 0.84 and Pi = 0.031, while the captive population was found to be Hd = 0.49 and Pi = 0.013 [41]. This indicates that in general, within a species, wild populations tend to be more diverse compared with captive populations and the Nubian ibex population in Oman shows the same pattern. Electronic supplementary material, table S6.3 in S6 shows additional comparisons of genetic diversity between the Nubian ibex and other species.

The haplotype networks for both the cytochrome *b* and D-loop, and the phylogenetic trees for the concatenated sequences illustrate a substantial differentiation between the Nubian ibex populations in the wild in Oman and those in captivity. The captive population haplotypes cluster with both the cytochrome *b* and D-loop haplotypes of the available NCBI reference sequences (electronic supplementary material S4, figures S4.2 and S4.3). This suggests that the source of these captive animals could be the Levant region or North Africa, as it is for the NCBI sequences with known geographical descriptors, given as Egypt [13] and the Dead Sea [14] (see electronic supplementary material S4, figures S4.2 and S4.3). The other NCBI sequences come from the Museum National d'Histoire Naturelle, France, and are likely to be from the European zoo population which is thought to have originated in North Africa [35]. It is unfortunate that more detailed descriptors of location are not available, and this illustrates the clear need for samples to be submitted to NCBI with more precise locality/origin descriptions.

There was no sharing of haplotypes between the wild Omani population and the captive animals at either mtDNA marker, which indicates that the Omani population is extremely distinctive from other Nubian ibex populations that have been studied and may deserve to be treated as a distinct taxonomic unit. This is supported by the AMOVA results, which showed a well-structured population with high variation among the populations (77.87% for D-loop and 97.19% for cytochrome *b*). The variance difference between the population is high ($F_{ST} = 0.77$ for D-loop and $F_{ST} = 0.97$ for cytochrome *b*). The distinctiveness of the Nubian ibex in Oman is also shown in the Bayesian phylogenetic trees of the concatenated sequences, which are supported by a high posterior probability of 1.0 (figure 6).

In terms of genetic divergence, our results are in the same range as several other studies, which investigated the taxonomic status of ungulates. For example, Wronski *et al.* [42] identified two different species of *Gazella*: *G. gazella* in the Levant and *G. arabica* in the Arabian Peninsula. This study estimated the average distance between these two populations as 12.7% for D-loop which is a similar level of divergence to our results (D-loop = 12.6%). Manceau *et al.* [43] identified an evolutionary significant unit (ESU) for one Pyrenean ibex (*C. pyrenaica*) population where the average distances between the Pyrenean and other Spanish populations were estimated to be D-loop = 5.3% and cytochrome *b* = 1.6%, and between the Pyrenean and the Alpine ibex to be D-loop = 5.7% and cytochrome *b* = 1.8%. Our results are higher than those of Manceau *et al.* [43] (D-loop = 12.6%; cytochrome *b* = 5.3%), which suggests strongly that the Nubian ibex in Oman may certainly deserve to be treated as a distinct

taxonomic unit. Whether it would be appropriate for this to be at the species or subspecies level requires additional sampling and nuclear data. We recognize that the capability of mitochondrial markers to specifically differentiate between species is limited due to introgression or incomplete lineage sorting [44].

In particular, the animals from captivity might not be reliable candidates for investigating the uniqueness of the Nubian ibex in Oman, because it has been documented that *Capra* species in captivity can hybridize with other species such as domestic goats [45]. Additionally, captive populations are susceptible to high levels of genetic drift due to founder effects. The founder history of the captive populations of the Nubian ibex is poorly known; however, it is unlikely to have a large founder base. For example, the breeding programme of the Association of Zoos and Aquariums in North America is thought to be founded from approximately 13 individuals [46]. Genetic drift could, therefore, exacerbate the divergence between the captive and wild populations.

Studies based on nuclear markers and samples from Saudi Arabia, Yemen, Jordan, Egypt and Sudan are now needed to ascertain the phylogeographic relationship between Nubian ibex populations across their range. However, this study sheds light on the putative difference between the wild Nubian ibex in Oman and its counterpart in the Levant and North Africa.

## 4.3. Gene flow and genetic diversity

The results from this study reveal that the population in the central region may be isolated and has lower genetic diversity than the southern populations, especially the Dhofar population. Small, isolated populations with restricted gene flow are at risk of inbreeding depression and further loss of genetic diversity. Inbreeding may cause reduction in reproductive fitness, and genetic diversity loss will reduce the ability of the population to adapt to changing environments [47]. Therefore, we recommend measures to limit the loss of genetic diversity. Wildlife corridors could be introduced and maintained between the Nubian ibex populations, or translocation of individuals between populations could replicate natural gene flow. The augmentation of an isolated, less genetically diverse population has been found to boost its genetic diversity and alleviate inbreeding depression [38,48]. Protection of the Nubian ibex is particularly important for Oman, specifically as it is considered one of the only a few large wild mammal species with appreciable free-ranging populations [17], and for the species as a whole, given its vulnerability to extinction [7], which may be greater than previously assessed, given the taxonomic questions raised by this study. If corridors between subpopulations are to be maintained, then protections need to be put in place to prevent their destruction from human activities, such as oil exploration, roads, hunting and agriculture. This could take the form of mitigation measures that combat and limit levels of disturbance, especially where there are oil exploration activities. Such developments must implement measures to reduce fragmentation and destruction of wildlife habitat and to mitigate the level of disturbance caused by infrastructure during development.

## 5. Conclusion

This study has demonstrated that the Nubian ibex population in Oman is highly distinct and not closely related to any of the assessed captive Nubian ibex populations in Oman and UAE. The mtDNA analysis of both cytochrome *b* and D-loop showed a deep divergence between the wild and the captive animals. Therefore, we recommend that captive animals are not used for any future reintroduction programmes until more detailed genetic data become available. If deemed appropriate and necessary, a captive-breeding programme in Oman should be initiated from wild individuals from southern and central regions, which should create a relatively diverse population that can be used for reinforcement to populations of low genetic diversity such as central region [49]. With this in mind, we currently recommend treating the wild Nubian ibex populations in Oman as distinct from captive populations and that they are managed separately. Their habitat must be conserved and protected from further human destruction, and monitoring strategies put in place that assess in relation to genetic diversity levels and population numbers through time.

Both translocation and captive-breeding strategies need to be carefully investigated prior to their inception to ensure that they do not harm existing wild populations, by following IUCN guidelines [49,50], and to ensure that they work in tandem with comprehensive *in situ* management of threats (e.g. the OnePlan approach) [51].

Further nuclear data and global reference samples covering the extant range of the Nubian ibex are required to elucidate the taxonomic position of the Nubian ibex in Oman and whether it deserves to be

treated as a distinct subspecies or species from Nubian ibex across the rest of Arabia and North Africa. This is not only important from an Oman conservation perspective, but is required to spread further light on population structure and connectivity across the range of Nubian ibex.

Ethics. Blood samples collected from captive centres in Oman and UAE were as part of routine veterinary examinations of captive-bred animals. From Oman, the approval for collecting and using blood samples of the Nubian ibex was taken from Dr Khalid Al-Rasbi director general of Bait Al Barakah breeding centre (the Diwan of the Royal Court, Muscat, Oman). Samples from UAE were used after approval from Mark Craig (Al Ain Zoo). The field work and samples collection were approved by the Office of Conservation for the Environment (Muscat, Oman) in 2014 as part of the Nubian ibex conservation programme.

Data accessibility. The datasets supporting this article have been uploaded as part of the electronic supplementary material. The sequences generated in this study were deposited to NCBI GenBank under accession numbers MW911255–MW911278.

Authors' contributions. M.K.A.-G. participated in the design of the study, collected the field samples, carried out the molecular laboratory work, analysed the data and drafted the manuscript; P.J.C.W. and R.A.B. participated in the design of the study and participated in reviewing the manuscript; K.L.D. and A.B developed the genetic analysis protocols and participated in reviewing the manuscript; M.G. participated in developing the genetics protocol and helped perform the molecular laboratory work; S.R. and T.A.-S. contributed to study deign and collected the field samples from the central region. M.A.-U. and H.A.-A. collected the field samples from the central region; H.A.-S., A.A., A.H. and N.Z. collected the field samples from the southern region; M.C. provided the samples from UAE captive centres; H.S. participated in the design of the study, developed the genetic analysis protocols and participated in reviewing the manuscript. All authors gave final approval for publication.

Competing interests. We declare we have no competing interests.

Funding. This project is funded and supported by the Diwan of the Royal Court and Office for Conservation of Environment, Sultanate of Oman.

Acknowledgements. We would like to express our sincere thanks to his excellency Sayyid Khalid bin Hilal Al-Busaidi the minister of the Diwan of the Royal Court, Oman, for his kind grant and support to this project. We would like to thank Mr Yasser Al-Salami, The Director General of Office for Conservation of Environment and Dr Mansoor Al-Jadhami for their continuous support to the project. Also a special thanks goes to Dr Qais Al-Rawahi and Salim Al-Rubaii for providing all the help and logistic support for collecting samples from central and southern regions, respectively. In addition, we would like to extend special thanks to Ms Hanan Al-Nabhani for her help in collecting samples from the Natural History Museum of Oman. We are extremely grateful to Balazs Buzas, Al Mayya Sanctuary, Fujairah, UAE, for the sharing of captive samples. Thanks also to David Mallon for comments on the draft.

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
