## [Peer Review File · Royal Society Open Science]

Review History

RSOS-210125.R0 (Original submission)

Review form: Reviewer 1 (Andrew Kitchener)

Is the manuscript scientifically sound in its present form?

Yes

Are the interpretations and conclusions justified by the results?

Yes

Is the language acceptable?

Yes

Do you have any ethical concerns with this paper?

No

Have you any concerns about statistical analyses in this paper?

No

Recommendation?

Accept with minor revision (please list in comments)

Comments to the Author(s)

I think this is a good and useful paper which requires minor revision. I would like to see some discussion of the taxonomic history of the Nubian ibex in the Arabian peninsula as there are candidate names for recognition of a possible new taxon in Oman/southern Arabia. An estimate of the divergence time between the two putative taxa would be useful and then trying to match this to the climatic and environmental history of the Arabian Peninsula would be useful. This would allow a prediction of where the split between populations of the two taxa may occur and provide a hypothesis and focus for future molecular and morphological studies.

I have made many minor comments on the paper (see Appendix A).

Review form: Reviewer 2 (Albano Beja-Pereira)**Is the manuscript scientifically sound in its present form?**

Yes

Are the interpretations and conclusions justified by the results?

No

Is the language acceptable?

Yes

Do you have any ethical concerns with this paper?

No

Have you any concerns about statistical analyses in this paper?

No

Recommendation?

Accept with minor revision (please list in comments)

Comments to the Author(s)

This is a great work on a species which very few data exists, particularly data from wild animals. The work is very complete, but I have some moderate/minor concerns that can be easily revised.

1. The authors shall provide the approximate date (year/season) of collection of the fecal samples. If possible, when comparing captive and wild populations it would be important to compare between the same year or close years. This would avoid generation overlap and its particular important given the founder effect that zoo/captive animals.

2. Mitochondrial genome is prone to drift (as it does not recombine and is haploid) and has a much higher mutation rate than the nuclear genome. Therefore, any inference regarding population connectivity and temporal divergence shall be made carefully. In this case, the authors while interpreting their data in terms of conservation did avoid going over the limits of mtDNA. As there is no male or nuclear genetics information, and it is well known that ungulates mostly disperse through males it is very difficult to infer isolation between populations. Therefore, the authors shall take this in consideration to reassess the discussion section of the MS. The same care shall be taken when discussing differentiation between captive and wild

populations. Captive populations drift very fast as they suffer from a founder effect event that can easily bias interpretations on population relationships. Of course, that "soft" recommendations shall be made based on these type of data, but this would require a larger population genetics study (to include nearly all Arabic Peninsula population) to have a larger understanding of the dynamics of this species. Although, I understand that this is not feasible (particular regarding Yemen) and these data merits publication.

Decision letter (RSOS-210125.R0)

Dear Mr Al Ghafri

On behalf of the Editors, we are pleased to inform you that your Manuscript RSOS-210125 "Genetic diversity of the Nubian ibex in Oman as revealed by mitochondrial DNA" has been accepted for publication in Royal Society Open Science subject to minor revision in accordance with the referees' reports. Please find the referees' comments along with any feedback from the Editors below my signature.

We invite you to respond to the comments and revise your manuscript. Please note that it will be important to carefully consider and take into account the points raised by both reviewers, in particular taking into account and revising your manuscript accordingly to deal with the important comments of Reviewer 2. Below the referees' and Editors' comments (where applicable) we provide additional requirements. Final acceptance of your manuscript is dependent on these requirements being met. We provide guidance below to help you prepare your revision.

Please submit your revised manuscript and required files (see below) no later than 7 days from today's (ie 07-Apr-2021) date. Note: the ScholarOne system will 'lock' if submission of the revision is attempted 7 or more days after the deadline. If you do not think you will be able to meet this deadline please contact the editorial office immediately.

on behalf of Professor Steve Brown (Subject Editor)
openscience@royalsociety.org

Reviewer comments to Author:

Reviewer: 1

Comments to the Author(s)

I think this is a good and useful paper which requires minor revision. I would like to see some discussion of the taxonomic history of the Nubian ibex in the Arabian peninsula as there are candidate names for recognition of a possible new taxon in Oman/southern Arabia. An estimate of the divergence time between the two putative taxa would be useful and then trying to match this to the climatic and environmental history of the Arabian Peninsula would be useful. This would allow a prediction of where the split between populations of the two taxa may occur and provide a hypothesis and focus for future molecular and morphological studies.

I have made many minor comments on the paper.

Reviewer: 2

Comments to the Author(s)

This is a great work on a species which very few data exists, particularly data from wild animals. The work is very complete, but I have some moderate/minor concerns that can be easily revised.

1. The authors shall provide the approximate date (year/season) of collection of the fecal samples. If possible, when comparing captive and wild populations it would be important to compare between the same year or close years. This would avoid generation overlap and its particular important given the founder effect that zoo/captive animals.

2. Mitochondrial genome is prone to drift (as it does not recombine and is haploid) and has a much higher mutation rate than the nuclear genome. Therefore, any inference regarding population connectivity and temporal divergence shall be made carefully. In this case, the authors while interpreting their data in terms of conservation did avoid going over the limits of mtDNA. As there is no male or nuclear genetics information, and it is well known that ungulates mostly disperse through males it is very difficult to infer isolation between populations. Therefore, the authors shall take this in consideration to reassess the discussion section of the MS. The same care shall be taken when discussing differentiation between captive and wild populations. Captive populations drift very fast as they suffer from a founder effect event that can easily bias interpretations on population relationships. Of course, that "soft" recommendations shall be made based on these type of data, but this would require a larger population genetics study (to include nearly all Arabic Peninsula population) to have a larger understanding of the dynamics of this species. Although, I understand that this is not feasible (particular regarding Yemen) and these data merits publication.

===PREPARING YOUR MANUSCRIPT===

Please ensure that you include an acknowledgements' section before your reference list/bibliography. This should acknowledge anyone who assisted with your work, but does not

qualify as an author per the guidelines at <https://royalsociety.org/journals/ethics-policies/openness/>.

===PREPARING YOUR REVISION IN SCHOLARONE===

-- Ensure that your data access statement meets the requirements at <https://royalsociety.org/journals/authors/author-guidelines/#data>. You should ensure that you cite the dataset in your reference list. If you have deposited data etc in the Dryad repository, please only include the 'For publication' link at this stage. You should remove the 'For review' link.

Author's Response to Decision Letter for (RSOS-210125.R0)

See Appendix B.

Decision letter (RSOS-210125.R1)

Dear Mr Al Ghafri,

It is a pleasure to accept your manuscript entitled "Genetic diversity of the Nubian ibex in Oman as revealed by mitochondrial DNA" in its current form for publication in Royal Society Open Science.

on behalf of Professor Steve Brown (Subject Editor)
openscience@royalsociety.org

Appendix A**ROYAL SOCIETY
OPEN SCIENCE****Genetic diversity of the Nubian ibex in Oman as revealed by
mitochondrial DNA**

Journal:	Royal Society Open Science
Manuscript ID	RSOS-210125
Article Type:	Research
Date Submitted by the Author:	27-Jan-2021
Complete List of Authors:	Al Ghafri, Mataab; Edinburgh Napier University, School of Applied Sciences; Royal Zoological Society of Scotland, RZSS WildGenes; Office for the Conservation of Environment, Diwan of Royal Court White, Patrick ; Edinburgh Napier University, School of Applied Sciences Briers, Robert; Edinburgh Napier University, School of Life Sciences Dicks, Kara ; Royal Zoological Society of Scotland, RZSS WildGenes Ball, Alex; Royal Zoological Society of Scotland, RZSS WildGenes Ghazali, Muhammad; Royal Zoological Society of Scotland, RZSS WildGenes Ross, Steve; Office for Conservation of the Environment, Diwan of Royal Court Al-Said, Taimur; Office for the Conservation of Environment, Diwan of Royal Court Al-Amri, Haitham; Office for the Conservation of Environment, Diwan of Royal Court Al-Umairi, Mudhafar; Office for the Conservation of Environment, Diwan of Royal Court Al Saadi, Hani; Office for the Conservation of Environment, Diwan of Royal Court Aka'ak, Ali; Office for the Conservation of Environment, Diwan of Royal Court Hardan, Ahmed; Office for the Conservation of Environment, Diwan of Royal Court Zabanoot, Nasser; Office for the Conservation of Environment, Diwan of Royal Court Craig, Mark; Al Ain Zoo, Life Sciences Department Senn, Helen ; Royal Zoological Society of Scotland, RZSS WildGenes
Subject:	evolution < BIOLOGY, genetics < BIOLOGY, taxonomy and systematics < BIOLOGY
Keywords:	Genetic diversity, Nubian ibex, cytochrome b, D-loop, Oman, conservation
Subject Category:	Genetics and genomics

Author-supplied statements

Relevant information will appear here if provided.

Ethics

Does your article include research that required ethical approval or permits?:

Yes

Statement (if applicable):

Blood samples collected from captive centres in Oman and UAE were as part of routine veterinary examinations of captive bred animals. From Oman, the approval for collecting and using blood samples of the Nubian ibex was taken from Dr Khalid Al-Rasbi director general of Bait Al Barakah breeding centre (the Diwan of the Royal Court, Muscat, Oman). Samples from UAE were used after approval from Mark Craig (Al-Ain zoo).

The field work and samples collection were approved by the Office of Conservation for the Environment (Muscat, Oman) in 2014 as part of the Nubian ibex conservation program.

Data

It is a condition of publication that data, code and materials supporting your paper are made publicly available. Does your paper present new data?:

Yes

Statement (if applicable):

The datasets supporting this article have been uploaded as part of the supplementary material. The sequences generated in this study will be deposited on NCBI GenBank and the accession numbers will be provided later

The Haplotypes sequences data for Cytochrome b and D-loop can be accessed through this private link: <https://www.dropbox.com/sh/psx7o33xdokilg5/AADhmRhV9VmC7AyNGGIboFdx?dl=0>
There are two downloadable files containing sequences in FASTA format. The first one containing seven sequences representing the seven Cytochrome b haplotypes sequences. The second one contains 17 sequencing representing the 17 D-loop haplotypes. All sequences were products of this study.

Conflict of interest

I/We declare we have no competing interests

Statement (if applicable):

CUST_STATE_CONFLICT :No data available.

Authors' contributions

This paper has multiple authors and our individual contributions were as below

Statement (if applicable):

M.A. participated in the design of the study, collected the field samples, carried out the molecular laboratory work, participated in data analysis, carried out sequence alignments and drafted the manuscript; P.W and R.B participated in the design of the study and participated in drafting the

manuscript; K.D and A.B developed the genetic analysis protocols and participated in drafting the
manuscript; M.G. has participated in developing the genetics protocol and carried out the molecular
laboratory work; S.T, T.A. contributed in study deign and collected the field samples from the central
region. M.A and H.A collected the field samples from the central region; H.A, A.A, A.H and N.Z
collected the field samples from the southern region; M.C provided the samples from UAE captive
centres; H.S. participated in the design of the study, developed the genetic analysis protocols and
participated in drafting the manuscript. All authors gave final approval for publication

1 Genetic diversity of the Nubian ibex in Oman as revealed by 2 mitochondrial DNA

Mataab K. Al-Ghafri^{1,2,3}, Patrick White¹, Robert A Briers¹, Kara L Dicks², Alex, Ball²,
Muhammad Ghazali², Steven Ross³, Taimur Al-Said³, Haitham Al-Amri³, Mudhafar Al-Umairi³,
Hani Al Saadi³, Ali Aka'ak³, Ahmed Hardan³, Nasser Zabanoot³, Mark Craig⁴, Helen Senn²

1 Edinburgh Napier University, School of Applied Sciences, Edinburgh, United Kingdom

2 RZSS WildGenes, Royal Zoological Society of Scotland, Edinburgh, United Kingdom

3 Office for the Conservation of Environment, Diwan of Royal Court, Muscat, Oman

4 Al Ain Zoo, Life Sciences Department, Abu Dhabi, United Arab Emirates.

Abstract:

The Nubian ibex (*Capra nubiana*) is patchily distributed across parts of Africa and
Arabia. In Oman, the Nubian ibex is one of the few free ranging wild mammals
found in the central and southern regions. It is suffering from declining population
numbers due to habitat degradation, human settlement expansion, poaching,
~~isolation~~ and population fragmentation. In order to study the impact of these factors
on the Nubian ibex population in Oman and inform future management plans, we
investigated the population's genetic diversity using mitochondrial DNA
(cytochrome *b* 487bp and D-loop 186bp). We found that the Nubian ibex in the
southern region of Oman was more diverse (haplotype diversity for D-loop 0.838)
compared to the central region (0.511) and gene flow between them was restricted.
We compared the genetic profiles of wild Nubian ibex from Oman with captive ibex
from two centres: UAE and Oman. A Bayesian phylogenetic tree of concatenated
sequences of cytochrome *b* and D-loop showed that wild Nubian ibex form a distinct
clade independent from captive animals. This divergence was supported by high
mean distances (0.126 for D-loop and 0.0528 for cytochrome *b*) and high F_{ST}

statistics (D-loop 0.725 and cytochrome *b* 0.968). These results indicate that captive
ibex are highly unlikely to have originated from the wild Nubian ibex population in
Oman and the considerable divergence suggests that the wild population in Oman is
treated as a distinct taxonomic unit. The results have implications for the
conservation management of this species and we advise that current captive
populations of ibex should not be used for any future re-introduction program for
the Nubian ibex population in Oman, and that genetic screening is conducted prior
to any future conservation action. Further nuclear genetic work and wider sampling
will be required to fully elucidate the degree of global taxonomic divergence of
Nubian ibex populations.

Keywords: Genetic diversity, Nubian ibex, cytochrome *b*, D-loop, Oman,
conservation.

**Introduction**

The *Capra* (or goat) genus is distributed widely in the three continents of Europe,
Africa and Asia, and its range extends from the cooler areas of the Alpine mountains
to the hot hyper-arid desert of Arabia (1). The number of species of *Capra* is debated
and described as containing between six and nine species, but they are all rocky
montane specialists (2) (1). According to phenotypic characteristics, *Capra* species
are divided into three classes: markhor, ibex and true goats (3). Heptner et al. (1961)
divided the ibex into seven species: Spanish ibex (*C. pyrenaica*), Alpine ibex (*C.*
*ibex*), Dagestan tur (*C. cylindricornis*), Caucasian ibex (*C. caucasica*), Siberian ibex
(*C. sibirica*), Nubian ibex (*C. nubiana*) and Walie ibex (*C. walie*). The more widely
accepted classification, which is used by the International Union for the

Conservation of Nature (IUCN), is that the *Capra* consists of nine species: the seven
ibex, a single species of markhor (*C. falconeri*) and wild goat (*C. aegagrus*) (see
supplementary materials S1).
Nubian ibex (*C. nubiana*) is the smallest *Capra* species (3). Males weigh between
55 and 65 kg and are distinguished by long curved horns, while females are much
lighter, weighing in the region of 21 to 27 kg with smaller and thinner horns (5) (6)

[revised manuscript text omitted]

	NUB_CytB_RZSS_R2	GCAGGTCTGGGTGTGAATAGT	

	NUB_CytB_RZSS_F3	CTGCTCTTCCTCCACGAAAC	369bp
	NUB_CytB_RZSS_R3	TGGGCGGAATATTATGCTTC	

PCRs were carried out at a total volume of 10 uL and contained 1 μ M of each
 primer, 1.4X DreamTaq Hot Start Master Mix (Thermo Fisher Scientific Inc.,
 includes 0.28 mM of each dNTP and 2.8mM MgCl₂) and 1 μ L of extracted DNA.
 For the faecal samples, 0.2 uL of Bovine Serum Albumin (BSA) was added to
 the master mix. The PCR program for both D-loop and cytochrome *b* was as
 follows. The initial denaturation step was at 95°C for 5 minutes. This was
 followed by 39 cycles of 95°C denaturation for 30 seconds, 55°C for 30 seconds
 to allow primer annealing and then 72°C for 60 seconds for elongation. A final
 72°C extension for 10 minutes completed the program. PCR products were
 confirmed by gel electrophoresis before being cleaned up with 0.5 μ l of EXO1
 enzyme and 0.5 μ l FastAP. PCR products were Sanger sequenced in both
 directions using the Big Dye Terminator Kit v3.1 (Applied Biosystems) on an
 ABI 3730 DNA Analyzer (Applied Biosystems).

Sequence Analysis

Sanger sequences were trimmed and quality checked by eye using Geneious
 software (Version 11.1.5). The three overlapping cytochrome *b* sequences
 generated from the same sample were aligned and the consensus sequence
 generated. The good quality sequences were then aligned with reference
 sequences from GenBank (see supplementary materials S4) and trimmed to an
 equal size of 186bp (D-loop) and 486bp (cytochrome *b*) using MEGA X (23).

To assess the genetic diversity of the Nubian ibex from Oman in relation to
 individuals from other sources, median joining (24) and TCS haplotype networks

were built using PopArt software v1.7 (25) for both the D-loop and cytochrome
*b*. To estimate the robustness of the sample sizes obtained by this study we
produced haplotype accumulation curves using the R (v.3.5.3) package HACSim
(26–28). This is used to estimate the total sample size which is required to capture
all the haplotypes in a specific population.

The accession numbers for other *Capra* species used to construct the phylogenetic
trees can be found in the supplementary materials (S4). Concatenation of D-loop
and cytochrome *b* to a total length of 673bp was performed using Geneious
(Version 11.1.5). In addition, the cytochrome *b* sequences were translated into
amino acids to check for evidence of incorrect amplification of nuclear insertions
of mitochondrial sequences (NuMtS) (29)

**Phylogenetic analysis**

A phylogenetic tree was constructed for the concatenated sequences. Whole
mitochondrial genome sequences for different *Capra* species along with other
species sequences imported from GenBank were used and can be found in the
supplementary materials (S3). *Bos taurus* (AY676870) was used as an outgroup. The
evolutionary model used for conducting the analyses was HKY+ Γ +I, as selected
using jModelTest in the R package “phangorn” (30). The phylogenetic tree for the
concatenated sequences was constructed by MrBayes (31) within Geneious (version
11.1.5). The parameters used were as follows: total chain length 1,000,000,
subsample frequency 200, and a burn-in of 10% of the trees was applied.

**Genetic diversity statistics**

Mean distances between the wild Nubian ibex from each location in Oman and the
captive animals were calculated with MEGA X using the maximum composite
likelihood (23,32). In addition, the genetic distance and differentiation between the
wild Nubian ibex and the captive ibex were calculated using DnaSP v 6.12.03 (33).
Analysis of molecular variance (AMOVA) was carried out using Arlequin v3.5 (34).
This was used to measure the population genetic structure within and between
groups, using 159 and 131 sequences for D-loop and cytochrome *b*, respectively.
The permutation was set to default (1,023) at a significance value of $p=0.05$.

**Results**

**Haplotype networks:**

A total of 188 of 228 samples were successfully amplified for the cytochrome *b*. The
quality control check excluded 57 samples. The final sample set therefore included
131 high quality sequences, which were used to create an alignment with total length

of 487bp for the analysis (see supplementary materials S2.2). A total of seven
 cytochrome *b* haplotypes were found in this study (named, A to G). Wild Nubian
 **ibex** from Oman were found to have three haplotypes (A, B and C), while the captive
 populations were found to contain four haplotypes (D, E, F and G). There were no
 shared haplotypes between the wild and captive populations. Regarding the wild
 sampling locations in Oman, two haplotypes were identified in the WWR (A and B).
 Dhofar was found to share both these haplotypes while, Shalim only had one
 haplotype (A, shared with both WWR and Dhofar). Dhofar on the other hand had an
 additional unique haplotype (C). The **geographic** distribution of the haplotypes of
 the wild Nubian ibex in Oman is illustrated in (figure 2).

Figure 2. The distribution of the mitochondrial DNA haplotypes within each
 location. The lower circle for each region represents cytochrome *b*, while the upper
 circle is D-loop. The size of the circle does not represent sample size.

 The Omani captive population had three cytochrome *b* haplotypes in total, two
 unique haplotypes (D and G) and shared one haplotype with the UAE captive
 population (E). In addition, the UAE captive population had an additional haplotype
 (F; figure 3) (supplementary materials S2.3). All haplotype sequences were
 submitted to GenBank under accession numbers (xxxxxxx).(note to
 editors/reviewers: the accession numbers will be added as soon as the sequences are
 deposited to the NCBI) (the data can be accessed through this private link:
 <https://www.dropbox.com/sh/psx7o33xdokilg5/AADhmRhV9VmC7AyNGGIboFdxa?dl=0>)

263

1
2

[revised manuscript text omitted]

deposited on NCBI GenBank and the accession numbers will be provided later.
Haplotypes sequences data for Cytochrome *b* and Dloop can be accessed through
this private link:

<https://www.dropbox.com/sh/psx7o33xdokilg5/AADhmRhV9VmC7AyNGGIboFdx?dl=0>

*Ethics statement.* Blood samples collected from captive centres in Oman and UAE
were as part of routine veterinary examinations of captive bred animals. From
Oman, the approval for collecting and using blood samples of the Nubian ibex was
taken from Dr Khalid Al-Rasbi director general of Bait Al Barakah breeding
centre (the Diwan of the Royal Court, Muscat, Oman). Samples from UAE were
used after approval from Mark Craig (Al-Ain zoo).

The filed work and samples collection were approved by the Office of
Conservation for the Environment (Muscat, Oman) in 2014 as part of the Nubian
ibex conservation program.

*Authors' contributions.* M.A. participated in the design of the study, collected the
field samples, carried out the molecular laboratory work, participated in data
analysis, carried out sequence alignments and drafted the manuscript; P.W and R.B
participated in the design of the study and participated in drafting the manuscript;
602 K.D and A.B developed the genetic analysis protocols and participated in drafting
the manuscript; M.G. has participated in developing the genetics protocol and
carried out the molecular laboratory work; S.T, T.A. contributed in study deign and
collected the field samples from the central region. M.A and H.A collected the field
samples from the central region; H.A, A.A, A.H and N.Z collected the field samples
from the southern region; M.C provided the samples from UAE captive centres; H.S.

participated in the design of the study, developed the genetic analysis protocols and
participated in drafting the manuscript. All authors gave final approval for
publication

*Competing interests.* We have no competing interests

*Funding.* This project is funded and supported by the Diwan of the Royal Court and
Office for Conservation of Environment, Sultanate of Oman.

*Acknowledgements.* We would like to express our sincere thanks to his excellency
Sayyid Khalid bin Hilal al Busaidi the minister of the Diwan of the Royal Court,
Oman, for his kind grant and support to this project. We would like to thank Mr
Yasser Al Salami, The Director General of Office for Conservation of Environment
and Dr Mansoor Al-Jadhmi for their continuous support to the project. Also a
special thanks goes to Dr Qais Al-Rawahi and Salim Al-Rubaii for providing all the
help and logistic support for collecting samples from central and southern regions
respectively. In addition, we would like to extend special thanks to Ms Hanan Al-
Nabhani for her help in collecting samples from the Natural History Museum of
Oman. We are extremely grateful to Balazs Buzas, Al Mayya Sanctuary, Fujairah,
UAE for the sharing of captive samples. Thanks also David Mallon for comments
on the draft.

**References**

1. Shackleton D. Wild Sheep and Goats and their Relatives. Status Survey and Conservation
Action Plan for Caprinae. Gland IUCN, 1997. 1997;
2. Schaller GB. Mountain Monarchs. Chicago, London: The University of Chicago Press;

1977.
3. Groves C, Grubb P. Ungulate Taxonomy. Baltimore, Maryland: Johns Hopkins University
Press; 2011. 317 p.
4. Heptner VG, Nasimovich AA, Bannikov AG. Mlekopitayushchiye Sovetskogo Soyuz. *Pamokopytnyye i Neparnokopytnyye* [Mammals of the Soviet Union. Artiodactyls and
Pamokopytnyye i Neparnokopytnyye [Mammals of the Soviet Union. Artiodactyls and
perissodactyls]. Vyshaya Shkola Publ. 1961;1:1–771.
5. Massolo A, Spalton JA, Tear TH, Lawrence MW, Said Al Harsusi L, Lovari S. Dynamic
social system in Nubian ibex: Can a second mating season develop in response to arid
climate? *J Zool.* 2008;274(3):216–25.
6. Habibi K. Group dynamics of the Nubian ibex (*Capra ibex nubiana*) in the Tuwayiq
Canyons, Saudi Arabia. *J Zool.* 1997;241:791–801.
7. Alkon P., Harding L, Jdeidi T, Masseti M, Nader I, de Smet K, et al. *Capra nubiana*.
IUCN Red List Threat Species [Internet]. 2008; Available from:
<http://dx.doi.org/10.2305/IUCN.UK.2008.RLTS.T3796A10084254.EN>
8. Ross S, Alqamy H El, Alsaid T. *Capra nubiana*, Nubian Ibex. The IUCN Red List of
Threatened Species 2020. 2020;(July).
9. Stüwe M, Scribner KT, Alkon PU. A comparison of genetic diversity in Nubian ibex
(*Capra ibex nubiana*) and Alpine ibex (*Capra i. ibex*). *Z für Sougetierkd.* 1992;57:120–3.
10. Granjon L, Vassart M, Greth A. Genetic variability in nubian ibex. *Mammalia*.
1990;54:665–7.
11. Bibi F, Vrba E, Fack F. A new african fossil caprin and a combined molecular and
morphological bayesian phylogenetic analysis of caprini (Mammalia: Bovidae). *J Evol*
*Biol.* 2012;25(9):1843–54.
12. Kazanskaya EY, Kuznetsova M V., Danilkin AA. Phylogenetic reconstructions in the
genus *Capra* (Bovidae, Artiodactyla) based on the mitochondrial DNA analysis. *Russ J*
*Genet* [Internet]. 2007;43(2):181–9. Available from:
<http://link.springer.com/10.1134/S1022795407020135>
13. Manceau V, Després L, Bouvet J, Taberlet P. Systematics of the genus *Capra* inferred
from mitochondrial DNA sequence data. *Mol Phylogenet Evol.* 1999;13(3):504–10.
14. Pidancier N, Jordan S, Luikart G, Taberlet P. Evolutionary history of the genus *Capra*
(Mammalia, Artiodactyla): Discordance between mitochondrial DNA and Y-chromosome
phylogenies. *Mol Phylogenet Evol.* 2006;40(3):739–49.
15. Hartl GB, Burger H, Willing R, Suchentrunk F. On the biochemical systematics of the
Caprini and the Rupricaprini. *Biochem Syst Ecol.* 1990;18:175–182.
16. Ropiquet A, Hassanin A. Hybrid origin of the Pliocene ancestor of wild goats. *Mol*
*Phylogenet Evol.* 2006;41(2):395–404.
17. CBD. 5th National Report to the Convention on Biological Diversity (CBD): Ministry of
Environment and Climate Affairs [Internet]. 2014. Available from:

<https://www.cbd.int/reports/nr5/>
18. Grobler M. A field guide to the larger mammals of Oman [Internet]. Muscat: Ministry of
Regional Municipalities, Environment & Water Resources; 2002. 48+48. Available from:
www.mrmewr.gov.om
19. Patzelt A. Synopsis of the Flora and Vegetation of Oman, with Special Emphasis on
Patterns of Plant Endemism. Braunschweig Wissenschaftliche Gesellschaft. 2015 Jan 1;282–
317.
20. Oman T of. Three arrested for poaching in Oman - Times Of Oman. 2018 Jan 22 [cited
2019 Jan 28]; Available from: <https://timesofoman.com/article/126561>
21. Untergasser A, Nijveen H, Rao X, Bisseling T, Geurts R, Leunissen JAM. Primer3Plus, an
enhanced web interface to Primer3. *Nucleic Acids Res* [Internet]. 2007/05/07. 2007
Jul;35(Web Server issue):W71–4. Available from:
<https://pubmed.ncbi.nlm.nih.gov/17485472>
22. Lindahl T. Instability and decay of the primary structure of DNA. *Nature*. 1993;362:709–
715.
23. Kumar S, Stecher G, Li M, Knyaz C, Tamura K. MEGA X: Molecular evolutionary
genetics analysis across computing platforms. *Mol Biol Evol*. 2018;35(6):1547–9.
24. Bandelt HJ, Forster P, Röhl A. Median-joining networks for inferring intraspecific
phylogenies. *Mol Biol Evol* [Internet]. 1999 Jan 1;16(1):37–48. Available from:
<https://doi.org/10.1093/oxfordjournals.molbev.a026036>
25. Leigh J, Bryant D. PopART: Full-feature software for haplotype network construction.
*Methods. Ecol Evol* [Internet]. 2015 [cited 2019 Jan 13];6(9):1110–1116. Available from:
<http://popart.otago.ac.nz/index.shtml>
26. Phillips JD, French SH, Hanner RH, Gillis DJ. HACSIm: an R package to estimate
intraspecific sample sizes for genetic diversity assessment using haplotype accumulation
curves. *PeerJ Comput Sci*. 2020;6:e243.
27. Phillips JD, Gillis DJ, Hanner RH. Incomplete estimates of genetic diversity within
species: Implications for DNA barcoding. *Ecol Evol*. 2019;9(5):2996–3010.
28. Phillips JD, Gwiazdowski RA, Ashlock D, Hanner R. An exploration of sufficient
sampling effort to describe intraspecific DNA barcode haplotype diversity: examples from
the ray-finned fishes (Chordata: Actinopterygii). *DNA Barcodes*. 2016;3(1).
29. Hazkani-Covo E, Zeller RM, Martin W. Molecular poltergeists: Mitochondrial DNA
copies (numts) in sequenced nuclear genomes. *PLoS Genet*. 2010;6(2).
30. Schliep KP. phangorn: phylogenetic analysis in R. *Bioinformatics* [Internet]. 2010 Dec
17;27(4):592–3. Available from: <https://doi.org/10.1093/bioinformatics/btq706>
31. Huelsenbeck JP, Ronquist F. MRBAYES: Bayesian inference of phylogenetic trees .
*Bioinformatics* [Internet]. 2001 Aug 1;17(8):754–5. Available from:
<https://doi.org/10.1093/bioinformatics/17.8.754>

32. Tamura K, Nei M, Kumar S. Prospects for inferring very large phylogenies for inferring
very large phylogenies by using the neighbor-joining method. *Proc Natl Acad Sci U S A*
[Internet]. 2004 Jul 27;101(30):11030 LP – 11035. Available from:
<http://www.pnas.org/content/101/30/11030.abstract>
33. Rozas J, Ferrer-Mata A, Sánchez-DelBarrio JC, Guirao-Rico S, Librado P, Ramos-Onsins
SE, et al. DnaSP 6: DNA Sequence Polymorphism Analysis of Large Data Sets. *Mol Biol*
*Evol* [Internet]. 2017 Dec 1;34(12):3299–302. Available from:
<http://academic.oup.com/mbe/article/34/12/3299/4161815>
34. Excoffier L, Lischer HEL. Arlequin suite ver 3.5: a new series of programs to perform
population genetics analyses under Linux and Windows. *Mol Ecol Resour* [Internet]. 2010
May;10(3):564–7. Available from: [http://doi.wiley.com/10.1111/j.1755-](http://doi.wiley.com/10.1111/j.1755-0998.2010.02847.x)
[0998.2010.02847.x](http://doi.wiley.com/10.1111/j.1755-0998.2010.02847.x)
35. Hassanin A, Ropiquet A, Couloux A, Cruaud C. Evolution of the mitochondrial genome
in mammals living at high altitude: New insights from a study of the tribe Caprini
(Bovidae, Antilopinae). *J Mol Evol*. 2009;68(4):293–310.
36. Bouckaert R, Vaughan TG, Barido-Sottani J, Duchêne S, Fourment M, Gavryushkina A,
et al. BEAST 2.5: An advanced software platform for Bayesian evolutionary analysis.
*PLoS Comput Biol*. 2019;15(4):1–28.
37. Habibi K, Grainger J. Distribution and status of Nubian ibex in Saudi Arabia. *Oryx*.
1990;24(3):138–42.
38. Frankham R, Ballou JD, Ralls K, Eldridge M, Dubash MR, Fenster CB, et al. Genetic
Management of Fragmented Animal and Plant Populations [Internet]. 2017 [cited 2018
Jan 22]. 432 p. Available from: [https://www.oup.com.au/books/higher-](https://www.oup.com.au/books/higher-education/science/9780198783404-genetic-management-of-fragmented-animal-and-plant-populations)
[education/science/9780198783404-genetic-management-of-fragmented-animal-and-plant-](https://www.oup.com.au/books/higher-education/science/9780198783404-genetic-management-of-fragmented-animal-and-plant-populations)
[populations](https://www.oup.com.au/books/higher-education/science/9780198783404-genetic-management-of-fragmented-animal-and-plant-populations)
39. Ross S, Costanzi JM, Al Jahdhami M, Al Rawahi H, Ghazali M, Senn H. First evaluation
of the population structure, genetic diversity and landscape connectivity of the
Endangered Arabian tahr. *Mamm Biol* [Internet]. 2020;(0123456789). Available from:
<https://doi.org/10.1007/s42991-020-00072-4>
40. Spalton JA, Lawrence MW, Brend SA. Arabian oryx reintroduction in Oman: Successes
and setbacks. *Oryx*. 1999;33(2):168–75.
41. Senn H, Banfield L, Wachter T, Newby J, Rabeil T, Kaden J, et al. Splitting or lumping? A
conservation dilemma exemplified by the critically endangered dama gazelle (*Nanger*
*dama*). *PLoS One*. 2014;9(6).
42. Wronski T, Wachter T, Hammond RL, Winney B, Hundertmark KJ, Blacket MJ, et al.
Two reciprocally monophyletic mtDNA lineages elucidate the taxonomic status of
Mountain gazelles (*Gazella gazella*) Two reciprocally monophyletic mtDNA lineages
elucidate the taxonomic status of Mountain gazelles (*Gazella gazella*). 2010;2000.
43. Manceau V, Crampe JP, Boursot P, Taberlet P. Identification of evolutionary significant
units in the Spanish wild goat, *Capra pyrenaica* (Mammalia, Artiodactyla). *Anim Conserv*.

1999;2(1):33–9.
44. Funk DJ, Omland KE. Species-Level Paraphyly and Polyphyly: Frequency, Causes, and
Consequences, with Insights from Animal Mitochondrial DNA. *Annu Rev Ecol Evol Syst.*
2003;34:397–423.
45. Hammer SE, Schwammer HM, Suchentrunk F. Evidence for introgressive hybridization
of captive markhor (*Capra falconeri*) with domestic goat: Cautions for reintroduction.
*Biochem Genet.* 2008;46(3–4):216–26.
46. Frankham R, Ballou JD, Briscoe DA, McInnes KH. *A Primer of Conservation Genetics*
[Internet]. Cambridge University Press; 2004 [cited 2019 Jan 27]. 220 p. Available from:
<http://ebooks.cambridge.org/ref/id/CBO9780511817359>
47. Ralls K, Sunnucks P, Lacy RC, Frankham R. Genetic rescue: A critique of the evidence
supports maximizing genetic diversity rather than minimizing the introduction of
putatively harmful genetic variation. *Biol Conserv* [Internet]. 2020;251:108784. Available
from: <http://www.sciencedirect.com/science/article/pii/S0006320720308429>
48. IUCN/SSC. IUCN Species Survival Commission Guidelines on the Use of Ex Situ
Management for Species Conservation. Version 2. 2014;1–7. Available from:
<https://portals.iucn.org/library/sites/library/files/documents/2014-064.pdf>
49. IUCN/SSC. Guidelines for Reintroductions and Other Conservation
Translocations. Version 1.0. Gland, Switzerland: [Internet]. IUCN Species Survival
Commission. 2013. Available from: [https://www.iucn.org/content/guidelines-](https://www.iucn.org/content/guidelines-reintroductions-and-other-conservation-translocations)
[reintroductions-and-other-conservation-translocations](https://www.iucn.org/content/guidelines-reintroductions-and-other-conservation-translocations)
50. WAZA. Towards Integrated Species Conservation. Towar Integr Species Conserv
[Internet]. 2013;14(November):2–4. Available from:
http://www.waza.org/files/webcontent/1.public_site/5.conservation/integrated_species_co
[nservation/WAZA Magazine 14.pdf](http://www.waza.org/files/webcontent/1.public_site/5.conservation/integrated_species_co)

A

Figure 1. A) Nubian ibex male captured by camera trap at Al-Wusta Wildlife Reserve (WWR) (Oman) (by Steven Ross & Taimur Al-Said). B) distribution range of *C. nubiana* in the middle east and Africa (8). C) Map of Oman showing the sampling locations represented by black circles. The samples were collected from three different places: Al Wusta Wildlife Reserve (WWR) and surrounds, Shalim and Dhofar.

135x115mm (220 x 220 DPI)

Figure 1. A) Nubian ibex male captured by camera trap at Al-Wusta Wildlife Reserve (WWR) (Oman) (by Steven Ross & Taimur Al-Said). B) distribution range of *C. nubiana* in the middle east and Africa (8). C) Map of Oman showing the sampling locations represented by black circles. The samples were collected from three different places: Al Wusta Wildlife Reserve (WWR) and surrounds, Shalim and Dhofar.

436x199mm (150 x 150 DPI)

Figure 2. The distribution of the mitochondrial DNA haplotypes within each location. The lower circle for each region represents cytochrome b, while the upper circle is D-loop. The size of the circle does not represent sample size.

281x194mm (150 x 150 DPI)

Figure 3. Median joining haplotype network for cytochrome b (487bp). The wild animals from Oman are in the right-hand blue box, while the captive animals are in the left-hand orange box. Each coloured circle represents a single haplotype, the size of which is proportional to the number of samples. The tick marks represent the mutational steps between haplotypes. Three reference sequences of *C. nubiana* from GenBank were used.

279x149mm (150 x 150 DPI)

Figure 4. Median joining haplotype network for D-loop (186bp). The wild animals from Oman are in the right-hand blue box while the captive animals are in the left-hand orange box. Each coloured circle represents a single haplotype, the size of which is proportional to the number of samples. The tick marks represent the mutational steps between haplotypes.

279x144mm (150 x 150 DPI)

Figure 5. Haplotype accumulation curves for cytochrome b (left) and D-loop (right) within the wild Nubian ibex samples. For cytochrome b it is estimated that more than 98.5% of the haplotypes are represented, while for the D-loop this is 93.7%. The dashed lines show the number of haplotypes found corresponding with sampled individuals. The dotted lines represent the expected numbers of haplotypes which should be found in the given population.

215x134mm (150 x 150 DPI)

Figure 6. Bayesian phylogenetic tree of concatenated mtDNA sequences of cytochrome b (487bp) and D-loop (187bp) calculated by MrBayes. The blue box indicates Oman wild Nubian ibex (WildHAP1 to WildHAP13). The yellow box indicates the captive Nubian ibex (CaptiveHAP14 to CaptiveHAP18). GenBank sequences are indicated by their accession numbers adjacent to the species binomial. The numbers at the nodes represent the posterior probabilities.

222x190mm (150 x 150 DPI)

Appendix B

Response to decision letter

Dear Prof /Dr Andrew Dunn
Senior Publishing Editor
Royal Society Open Science,

Thank you for considering our manuscript “Genetic diversity of the Nubian ibex in Oman as revealed by mitochondrial DNA” in Royal Society Open Science. We appreciated the positive and constructive reviews. Please find below the comments made by the reviewers (in bold) and our responses below each.

Kind regards,

Mataab K. Al Ghafri

Reviewer: 1

Comments to the Author(s)

I think this is a good and useful paper which requires minor revision. I would like to see some discussion of the taxonomic history of the Nubian ibex in the Arabian Peninsula as there are candidate names for recognition of a possible new taxon in Oman/southern Arabia. An estimate of the divergence time between the two putative taxa would be useful and then trying to match this to the climatic and environmental history of the Arabian Peninsula would be useful. This would allow a prediction of where the split between populations of the two taxa may occur and provide a hypothesis and focus for future molecular and morphological studies.

I have made many minor comments on the paper.

To answer these comments;

Point 1:

Unfortunately, there has previously been little to no taxonomic studies of the species within the Arabian Peninsula, due to a lack of samples. However, Harrison

(1) mentioned several taxonomies for the Nubian ibex in north east Africa and Arabia shown in the following table

taxa	details	locality
Capra nubiana	1825 F.Cuvier, in:Geoffroy &Cuvier, Histoires naturelles des Mammiferes,50:20	Nubia
Capra sinaitica	1833 Ehernberg, In: Hemprich & Ehernberg, Symbolae Physicae Mammalium, 2:18	Sinai
Capra Arabica	1835 Ruppell, Neue Wirbelthiere Abyssinien, Saugethiere: 17	Sinai
Aegoceros beden	1835 Wagner, Die Saugethiere in Abbildungen nach der Natur, 5:1303	Hejaz, Arabia
Capra mengesi	1896 Noak, Zoologischer Anz., 19:353	Hadramaut, south-eastern Arabia

On the other hand, Habibi in his book (2) did not provide enough information about the taxonomy of the Nubian ibex and did not discuss any putative differences between Nubian ibex species in Arabia.

So we can conclude that the inability to discuss the taxonomy of the Nubian ibex in Arabia can be attributed to the lack of studies, maybe due to the location of Oman and Yemen in the south eastern coast of the Arabian Peninsula which made them rarely visited by explorers.

To address this comment the following modified sentence has been added:

“However, a lack of specimens from southern Arabia has hindered the resolution of this debate and prevented further understanding of population sub-structure.”
Now in lines 83-85 (in tracked and clean version of the manuscript)

Point 2:

- Attempting to calibrate the divergence time, would be extremely useful, but is limited in this study due to the use of small fragments of mitochondrial data (cytochrome *b* and D-loop) and therefore the results would be

unreliable. The estimation of divergence time ideally requires fossil samples, several genes/full mitochondrion sequences and/or nuclear data.

Point 3:

- All spelling/grammar comments have been addressed.
- The reviewer preferred to use 'ibexes' as the plural for ibex. However, the word ibex can be used as both singular and plural and this is very common in the literature. Therefore, we prefer to use the word 'ibex' to be more consistent with the wider literature.

Reviewer: 2

Comments to the Author(s)

This is a great work on a species which very few data exists, particularly data from wild animals. The work is very complete, but I have some moderate/minor concerns that can be easily revised.

1. The authors shall provide the approximate date (year/season) of collection of the faecal samples. If possible, when comparing captive and wild populations it would be important to compare between the same year or close years. This would avoid generation overlap and its particular important given the founder effect that zoo/captive animals.

To answer these comments:

Collection dates has been inserted in table 2 and the following sentence has been added "Faecal samples from the wild and blood samples from captive centres were collected between 2014 and 2018 (table 2)" – lines 153-155 in tracked version and lines 152-154 in clean version

2. Mitochondrial genome is prone to drift (as it does not recombine and is haploid) and has a much higher mutation rate than the nuclear genome. Therefore, any inference regarding population connectivity and temporal divergence shall be made carefully. In this case, the authors while interpreting their data in terms of conservation did avoid going over the limits of mtDNA. As there is no male or nuclear genetics information, and it is well known that ungulates mostly disperse through males it is very difficult to infer isolation

between populations. Therefore, the authors shall take this in consideration to reassess the discussion section of the MS. The same care shall be taken when discussing differentiation between captive and wild populations. Captive populations drift very fast as they suffer from a founder effect event that can easily bias interpretations on population relationships. Of course, that "soft" recommendations shall be made based on these type of data, but this would require a larger population genetics study (to include nearly all Arabic Peninsula population) to have a larger understanding of the dynamics of this species. Although, I understand that this is not feasible (particular regarding Yemen) and these data merits publication.

To answer these comments:

The following paragraphs has been added:

“It should be noted that the analyses carried out here have used only mitochondrial DNA, which is only inherited maternally, and are therefore limited in their scope to elucidate population structure and gene flow in a species for which male-biased dispersal is typical. Further research using nuclear makers would provide valuable insight into whether gene-flow between populations in Oman remains limited after accounting for male-biased dispersal” - lines 474-479 in tracked version and lines 468-473 in clean version

“Additionally, captive populations are susceptible to high levels of genetic drift due to founder effects. The founder history of the captive populations of the Nubian ibex is poorly known, however it is unlikely to have a large founder base. For example the breeding programme of the Association of Zoos and Aquariums in North America is thought to be founded from approximately 13 individuals (3). Genetic drift could, therefore, exacerbate the divergence between the captive and wild populations” –lines 545-551 in tracked version and 539-545 in clean version

The following sentences have been amended:

- 1- “The animals from captivity might not be reliable candidates for investigating the uniqueness of the Nubian ibex in Oman because it has been documented that *Capra* species in captivity can hybridize with other species such as domestic goats (45)”.

To be “In particular, the animals from captivity might not be reliable candidates for investigating the uniqueness of the Nubian ibex in Oman, because it has been documented that *Capra* species in captivity can hybridize with other species such as domestic goats (45)” now in lines 542-545 in tracked version and 536-539 in clean version

2- “However, this study sheds light on the putative difference between wild Nubian ibex in Oman and its counterpart in the Levant and North Africa” now in lines 554-556 in tracked version and lines 548-550 in clean version

References:

1. Harrison DL. The Mammals of Arabia. 2nd ed. Bates PJ, editor. Harrison Zoological Museum; 1991. 180–183 p.
2. Habibi K. The Desert Ibex; Life history, ecology and behaviour of the Nubian ibex in Saudi Arabia. first. NCWCD Saudi Arabia; 1994. 192 p.
3. Putnam AS, Nguyen TN, Mott A, Korody ML, Ryder OA. Assessing possible hybridization among managed Nubian ibex in North America. *Zoo Biol.* 2020;39(2):121–8.